# LEARNING TO GENERATE FILTERS FOR CONVOLUTIONAL NEURAL NETWORKS

## ABSTRACT

Conventionally, convolutional neural networks (CNNs) process different images with the same set of filters. However, the variations in images pose a challenge to this fashion. In this paper, we propose to generate sample-specific filters for convolutional layers in the forward pass. Since the filters are generated on-the-fly, the model becomes more flexible and can better fit the training data compared to traditional CNNs. In order to obtain sample-specific features, we extract the intermediate feature maps from an autoencoder. As filters are usually high dimensional, we propose to learn a set of coefficients instead of a set of filters. These coefficients are used to linearly combine the base filters from a filter repository to generate the final filters for a CNN. The proposed method is evaluated on MNIST, MTFL and CIFAR10 datasets. Experiment results demonstrate that the classification accuracy of the baseline model can be improved by using the proposed filter generation method.

## 1 INTRODUCTION

Variations exist widely in images. For example, in face images, faces present with different head poses and different illuminations which are challenges to most face recognition models. In the conventional training process of CNNs, filters are optimized to deal with different variations. The number of filters increases if more variations are added to the input data. However, for a test image, only a small number of the neurons in the network are activated which indicates inefficient computation (Xiong et al. (2015)).

Unlike CNNs with fixed filters, CNNs with dynamically generated sample-specific filters are more flexible since each input image is associated with a unique set of filters. Therefore, it provides possibility for the model to deal with variations without increasing model size.

However, there are two challenges for training CNNs with dynamic filter generation. The first challenge is how to learn sample-specific features for filter generation. Intuitively, filter sets should correspond to variations in images. If the factors of variations are restricted to some known factors such as face pose or illumination, we can use the prior knowledge to train a network to represent the variation as a feature vector. The main difficulty is that besides the factors of variations that we have already known, there are also a number of them that we are not aware of. Therefore, it is difficult to enumerate all the factors of variations and learn the mapping in a supervised manner. The second challenge is that how to map a feature vector to a set of new filters. Due to the high dimension of the filters, a direct mapping needs a large number of parameters which can be infeasible in real applications.

In response, we propose to use an autoencoder for variation representation leaning. Since the objective of an autoencoder is to reconstruct the input images from internal feature representations, each layer of the encoder contains sufficient information about the input image. Therefore, we extract features from each layer in the encoder as sample-specific features. For the generation of filters , given a sample-specific feature vector, we firstly construct a filter repository. Then we learn a matrix that maps the feature vector to a set of coefficients which will be used to linearly combine the base filters in the repository to generate new filters.

Our model has several elements of interest. Firstly, our model bridges the gap between the autoencoder network and the prediction network by mapping the autoencoder features to the filters in the

prediction network. Therefore, we embed the knowledge from unsupervised learning to supervised learning. Secondly, instead of generating new filters directly from a feature vector, we facilitate the generation with a filter repository which stores a small number of base filters. Thirdly, we use linear combination of the base filters in the repository to generate new filters. It can be easily implemented as a convolution operation so that the whole pipeline is differentiable with respect to the model parameters.

## 2 RELATED WORK

The essential part of the proposed method is the dynamical change of the parameters of a CNN. In general, there are two ways to achieve the goal including dynamically changing the connection and dynamically generating the weights, both of which are related to our work. In this section, we will give a brief review of the works from these two aspects.

### 2.1 DYNAMIC CONNECTION

There are several works in which only a subset of the connections in a CNN are activated in a forward pass. We term this kind of strategy dynamic connection. Since the activation of connections depends on input images, researchers try to find an efficient way to select subsets of the connections. The benefit of using dynamical connection is the reduction in computation cost.

Xiong et al. (2015) propose a conditional convlutional neural network to handle multimodal face recognition. They incorporate decision trees to dynamically select the connections so that images from different modalities activate different routes.

Kontschieder et al. (2015) present deep neural decision forests that unify classification trees with representation learning functionality. Each node of the tree performs routing decisions via a decision function. For each route, the input images are passed through a specific set of convolutional layers.

Ioannou et al. (2016) and Baek et al. (2017) also propose similar frameworks for combining decision forests and deep CNNs. Those hybrid models fuse the high representation learning capability of CNNs and the computation efficiency of decision trees.

### 2.2 DYNAMIC WEIGHT

We refer to weights that are dynamically generated as dynamic weights. Furthermore, since the weights are the parameters of a CNN, learning to generate those weights can also be viewed as a meta learning approach.

Bertinetto et al. (2016) propose to use dynamic weights in the scenario of one-shot learning. They construct a learnet to generate the weights of another deep model from a single exemplar. A number of factorizations of the parameters are proposed to reduce the learning difficulty.

Ha et al. (2016) present hypernetworks which can also generate weights for another network, especially a deep convolutional network or a long recurrent network. The hypernetworks can generate non-shared weights for LSTM and improve its capability of sequence modelling.

There are several other similar architectures (Schmidhuber (2008), De Brabandere et al. (2016)). Results from those works demonstrate that dynamical weights help learn feature representation more effectively.

The work that most resembles ours is the work of De Brabandere et al. (2016). However, our work is different in the following aspects. (i) The feature vectors we used for filter generation are extracted from the feature maps of an autoencoder network. (ii) New filters are generated by the linear combination of base filters in a filter repository.

The rest of the paper is structured as follows. Section 3 presents the details of the proposed method. Section 4 shows the experiment results and Section 5 concludes the paper.

## 3 METHOD

The framework of the proposed method is illustrated in Figure 1. The description of our model will be divided into three parts, i.e. sample-specific feature learning, filter generation, and final prediction.

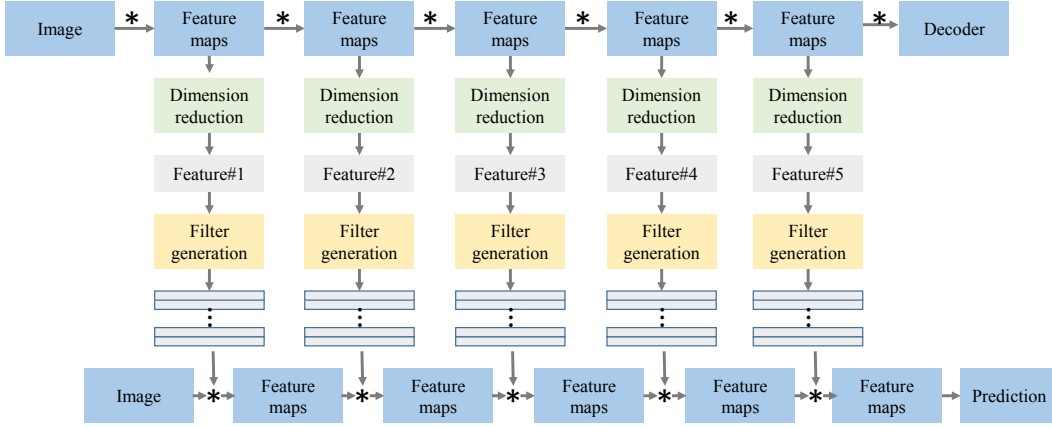

Figure 1: The framework of the proposed method. The autoencoder network in the first row is used to extract features from the input image. The obtained feature maps are fed to a dimension reduction module to reduce the dimension of the feature maps. Then the reduced features are used to generate new filters in the filter generation module. Finally, the prediction network takes in the same input image and the generated filters to make the final prediction for high level tasks such as detection, classification and so on. "*" indicates the convolution operation.

### 3.1 SAMPLE-SPECIFIC FEATURE LEARNING

It is difficult to quantify variations in an image sample. Thus, we adopt an autoencoder to learn sample-specific features. Typically, an autoender consists of an encoder and a decoder. The encoder extracts features from the input data layer by layer while the decoder plays the role of image reconstruction. Therefore, we use the features from each layer of the encoder as representations of the input image. Since the feature maps from the encoder are three-dimensional, we use dimension reduction modules to reduce the dimension of the feature maps. For each dimension reduction module, there are several convolutional layers with stride larger than 1 to reduce the spatial size of the feature maps to $1 \times 1$. After dimension reduction, we obtained the sample-specific features at different levels. The loss function for the autoencoder network is the binary cross entropy loss

$$L_{rec} = -1/N_{pix} \sum_i (t_i * log(o_i) + (1 - t_i) * log(1 - o_i)). \tag{1}$$

$N_{pix}$ is the number of pixels in the image. $o_i$ is the value of the $i$th element in the reconstructed image and $t_i$ is the value of the $i$th element in the input image. Both the input image and the output image are normalized to $[0, 1]$.

### 3.2 FILTER GENERATION

The filter generation process is shown in Figure 2. The input to the filter generation module is the sample-specific feature vector and the output is a set of the generated filters. If we ignore the bias term, a filter can be flatten to a vector. Given an input feature vector, the naive way to generate filters is to use a fully connected layer to directly map the input vector to the filters. However, it is infeasible when the number of filters is large. Let the length of each filter be $L_k$ and the length of the feature vector be $L_f$. If we need to generate $N$ filter vectors from the feature vector. We need $N \times L_f \times L_k$ parameters in a transformation matrix. $N \times L_k$ can be larger than ten thousand.

In order to tackle the problem, we refactor each filter vector $k_i$ as

$$k_i = \Sigma_{j=1}^{M}(w_j b_j). \tag{2}$$

$w_j$ is the coefficient of the base filter $b_j$ which is from a filter repository. $M$ is the number of filters in the filter repository. Equation 2 assumes that each filter vector can be generated by a set of base filters. The assumption holds true if $M = L_K$ and those base filters are orthogonal. However, in real applications of CNNs, each convolutional layer has limited number of filters which indicates that compared to the large dimension of the filter vector space, only a small subspace is used in the final trained model. Based on this observation, we set $M << L_k$ in this work. The total number of parameters in the transformation matrix is $N \times L_f \times M$ which is much smaller than the original size. The filters in the repositories are orthogonally initialized and optimized during the training process.

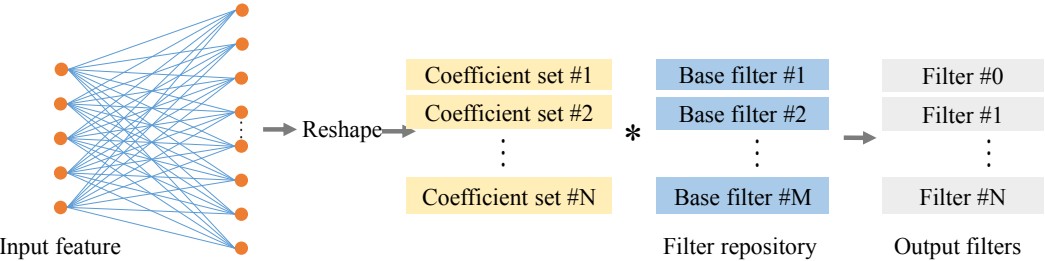

Figure 2: Filter generation process. The input feature vector is fed to a fully connected layer and the layer outputs sets of coefficients. Those coefficients linearly combine the base filters from a filter repository and generate new filters.

### 3.3 FINAL PREDICTION

The prediction network is the network for the high level task, such as image classification, recognition, detection and so on. The filters used in the prediction network are provided by the filter generation module while the weights of the classifier in the prediction network are learned during the training process.

Loss functions for high level tasks are task-dependent. In this work, we will use classification task for demonstration and the loss is the negative log likelihood loss

$$L_{cls} = -log(p_t), \tag{3}$$

where $t$ is the image label and $p_t$ is the softmax probability of the $t$th label.

Therefore, the entire loss function for training our model is

$$L = L_{rec} + L_{cls}. \tag{4}$$

## 4 EXPERIMENTS

The proposed method aims for generating dynamic filters to deal with variations and improve the performance of a baseline network. In the following experiments, we evaluate our method on three tasks, i.e. digit classification on MNIST dataset(Section 4.1), facial landmark detection on MTFL dataset (Section 4.1) and image classification on CIFAR10 dataset (Section 4.3). The number of the base filters in each filter repository is the same as the number of the filters in each layer of the prediction network if not specified. We will also present further analysis on the generated filters in Section 4.4. Details of all network structures are given in Appendix A.1.

### 4.1 MNIST EXPERIMENT

To begin our evaluation, we firstly set up a simple experiment on digit classification using MNIST dataset (LeCun et al. (1998)). We will show the accuracy improvement brought by our dynamic filters by comparing the performance difference of a baseline network with and without our dynamic filters. We will also analyze how the size of the encoder network and the size of the filter repository (the number of filters in the repository) effect the accuracy of digit classification.

Table 1: Classification accuracy on MNIST test set with $s = 5$.

|  | baseline | $n_{enc} = 5$ | $n_{enc} = 10$ | $n_{enc} = 20$ |
|---|---|---|---|---|
| acc@epoch=1 | 95.23% | 95.85% | 95.94% | 97.53% |
| acc@epoch=20 | 98.25% | 98.34% | 98.45% | 99.10% |

Table 2: Classification accuracy on MNIST test set with $n_{enc} = 20$.

|  | baseline | $s = 2$ | $s = 5$ | $s = 10$ |
|---|---|---|---|---|
| acc@epoch=1 | 95.23% | 97.11% | 97.53% | 97.29% |
| acc@epoch=20 | 98.25% | 98.55% | 99.10% | 98.92% |

### 4.1.1 SETTINGS

The baseline model used in this experiment is a small network with two convolutional layers followed by a fully connected layer that outputs ten dimensions. For simplicity, we only use five filters in each convolutional layer. Details of the network structures are shown in Appendix A.1.1. To evaluate the effect of the size of the encoder network, we compare the classification accuracy obtained when the encoder network has different numbers of filters in each layer. Let $n_{enc}$ be the number of filters in each layer of the encoder network. We choose $n_{enc}$ from $\{5, 10, 20\}$. We also choose different repository size $s$ from $\{2, 5, 10\}$. In the evaluation of the effect of $s$, we fix $n_{enc} = 20$ and we fix $s = 5$ to evaluate the effect of $n_{enc}$. We train this baseline model with and without filter generation for 20 epochs respectively.

### 4.1.2 RESULTS

We show the classification accuracy on the test set in Table 1 and Table 2. The first row shows the test accuracy after training the network for only one epoch and the second row shows the final test accuracy. From both tables, we can find that the final test accuracy of the baseline model using our dynamically generated filters is higher than that using fixed filters. The highest accuracy obtained by our generated filters is 99.1% while the accuracy of the fixed filters is 98.25%.

Interestingly, the test accuracies after the first epoch (first row in Table 1 and Table 2) show that our dynamically generated filters help the network fit the data better than the original baseline model. It could be attribute to the flexibility of the generated filters. Though there are only a small number of base filters in the repository, linear combination of those base filters can provide filters that efficiently extract discriminative features from input images.

In Table 1, when $s = 5$, the classification accuracy increases as encoder network has more filters. It is straightforward because with more filters, the encoder network can better capture the variation in the input image. So it can provide more information for the generation of filters. Based on the observation from Table 2, it seems that the final classification accuracy is less dependent on the repository size given $n_{enc} = 20$.

## 4.2 MTFL EXPERIMENT

In this section, we apply our filter generation to the task of facial landmark detection. To give a more straightforward understanding of the usefulness of our filter generation, we firstly investigate the performance difference of a baseline model before and after some explicit variations are added to the dataset. Then we show the detection performance improvement with respect to the size of the detection network.

### 4.2.1 SETTINGS

**Dataset.** MTFL dataset (Zhang et al. (2014)) contains 10,000 face images with ground truth landmark locations. In order to compare the performance difference of baseline models with respect to

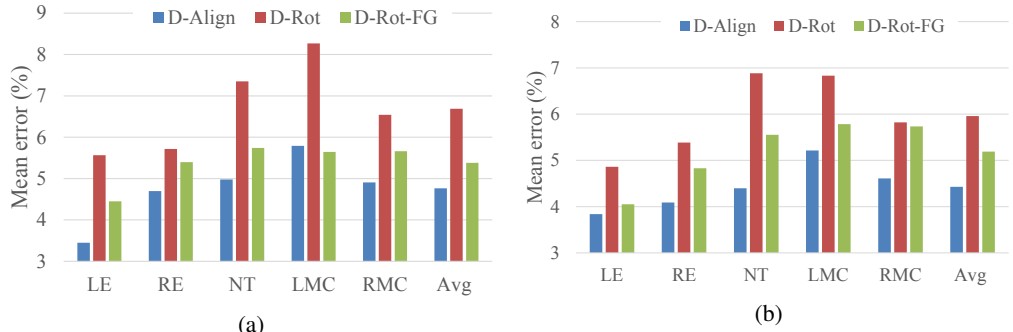

Figure 3: Facial landmark detection results on MTFL dataset. *D-Rot-FG* indicates the baseline model trained on *D-Rot* with our generated filters. LE means left eye. RE means right eye. NT means nose tip. LMC means left mouth center. RMC means right mouth center. AVG means the averaged mean error across all five landmarks. (a) Facial landmark detection using baseline model $Model_{32}$.(b) Facial landmark detection using baseline model $Model_{64}$.

variations, we construct two datasets from the original MTFL dataset. Rotation variation is used here since it can be easily introduced to the images by manually rotating the face images.

Dataset *D-Align*. We follow Yi et al. (2014) to aligned all face images and crop the face region to the size of $64 \times 64$.

Dataset *D-Rot*. This dataset is constructed based on *D-Align*. we randomly rotate all face images within $[-45°, 45°]$.

Some image samples for both datasets are shown in Appendix A.2 Figure 6.

We split both datasets into the training dataset containing 9,000 images and the test dataset containing 1,000 images. Note that the train-test splits in *D-Align* and *D-Rot* are identical.

**Models** Here we train two baseline models based on UNet(Ronneberger et al. (2015)). The baseline models are $Model_{32}$ with 32 filters in each convolutional layer and $Model_{64}$ with 64 filters in each convolutional layer. $Model_{32}$ and $Model_{64}$ share the same architecture. Details of the network structures are shown in Appendix A.1.2.

We firstly trained $Model_{32}$ and $Model_{64}$ on *D-Align* and *D-Rot* without our filter generation module. Then we train them on *D-Rot* with our filter generation module. For evaluation, we use two metrics here. One is the mean error which is defined as the *average* landmark distance to ground-truth, normalized as percentages with respect to interocular distance (Burgos-Artizzu et al. (2013)). The other is the *maximal* normalized landmark distance.

### 4.2.2 RESULTS

Since there are more rotation variations in *D-Rot* than *D-Align*, we can consider landmark detection task on *D-Rot* is more challenging than that on *D-Align*. This is also proved by the increase in detection error when the dataset is switched from *D-Align* to *D-Rot* as shown in Figure 3 and Table 3. However, when we train the same baseline model with our generated filters, the detection error decreases, compared to the same model trained on *D-Rot*. There is also a large error drop in maximal detection error. These results indicate that using filters conditioned on the input image can reduces the effect of variations in the dataset.

Comparing the averaged mean error in Figure 3a and Figure 3b, we find that the performance gain brought by filter generation is larger on $Model_{32}$ than that on $Model_{64}$. It could be explained by the capacity of the baseline models. The capacity of $Model_{64}$ is larger than that of $Model_{32}$. So $Model_{64}$ can handle more variations than $Model_{32}$, so the performance gain on $Model_{64}$ is smaller.

Table 3: Maximal facial landmark detection error.

|  | D-Align | D-Rot | D-Rot-FG |
|---|---|---|---|
| $Model_{32}$ | 49.11% | 111.17% | 68.60% |
| $Model_{64}$ | 58.41% | 105.01% | 46.65% |

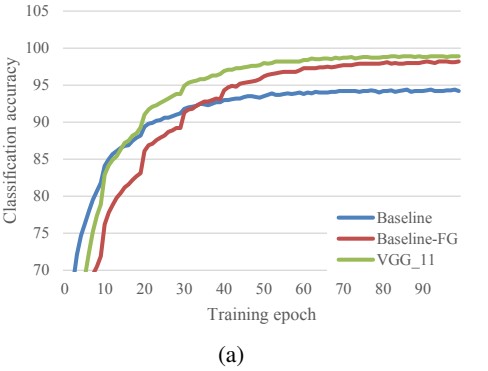 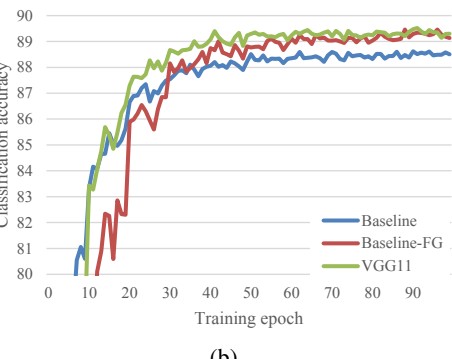

(a) (b)

Figure 4: Training accuracy (a) and test accuracy (b) on CIFAR10. "Baseline-FG" indicates the baseline model trained with our dynamic filter generation process.

### 4.3 CIFAR10 EXPERIMENT

CIFAR10(Krizhevsky & Hinton (2009)) dataset consists of natural images with more variations. We evaluate models on this dataset to show the effectiveness of our dynamic filter generation in this challenging scenario.

#### 4.3.1 SETTINGS

We construct a small baseline network with only four convolutional layers followed by a fully connected layer. We train this model on CIFAR10 firstly without filter generation and then with filter generation. We also train a VGG11 model (Simonyan & Zisserman (2014)) on this dataset.

#### 4.3.2 RESULTS

The results are shown in Figure 4. From the training accuracy curves, we observe that the baseline model trained without filter generation doesn't fit the data as well as other models. This is because there are only five layers in the network which limits the network's capacity. When the baseline model is trained with filter generation, the model can fit the data well, reaching more than 98% training accuracy. VGG11 also achieves high training accuracy which is not supervising since there are more layers (eleven layers) in the models.

The test accuracy curves also show the benefit of adopting our dynamic filter generation. The baseline classification accuracy is improved by ∼1% by using filter generation and the test accuray is comparable to VGG11.

Based on the above evaluations on different datasets, we claim that dynamically generated filters can help improve the performance of the baseline models. Using linear combination of base filters from filter repositories can generate effective filters for high level tasks.

### 4.4 FILTER ANALYSIS

In this section, we visualize the distributions of the coefficients, the generated filters and the feature maps using MNIST dataset. Then we conduct another experiment on CIFAR10 dataset to demonstrate that the generated filters are sample-specific.

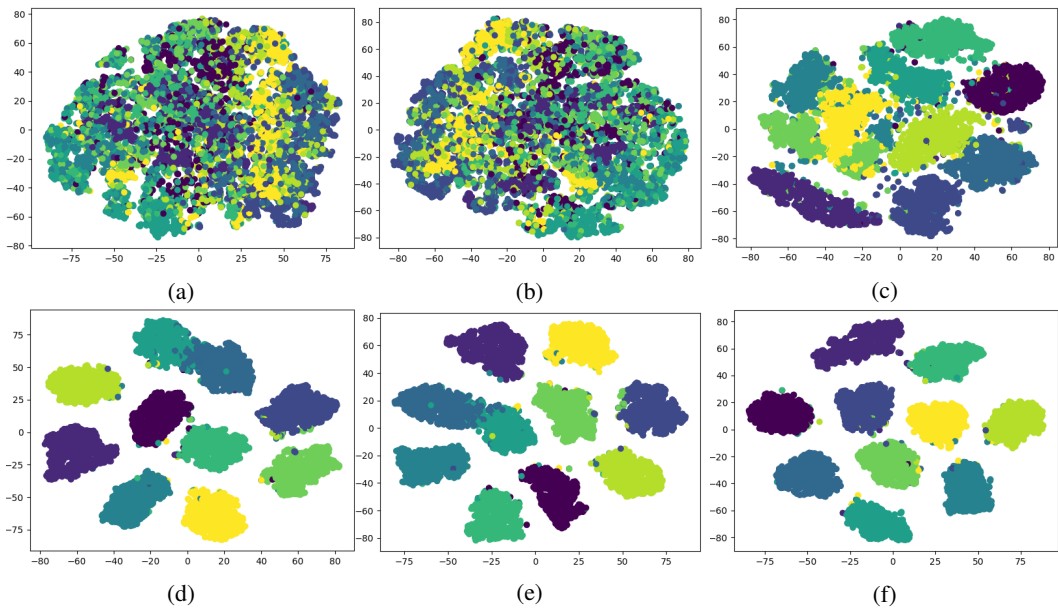

Figure 5: Visualization of the distributions of the generated coefficients, filters, and feature maps from the first (top row) and the second (bottom row) convolutional layer.

#### 4.4.1 DISTRIBUTION VISUALIZATION

The model we used for visualization is the baseline model trained in MNIST experiment with $n_{ae} = 20$ and $s = 5$. TSNE (Maaten & Hinton (2008)) is applied to project the high dimensional features into a two-dimensional space. The visualization results are shown in Figure 5. In the first row, we show the distributions of the coefficients, the filters, and the feature maps from the first convolutional layer of the model. We observe that the generated filters are shared by certain categories but not all the categories. It is clear in Figure 5a that the coefficients generated by some digits are far away from those by other digits. Nevertheless, the feature maps from the first convolution layer show some separability. In the second row, the generated coefficients and the generated filters forms into clusters which means digits from different categories activate different filters. This behavior makes the final feature maps more separable.

#### 4.4.2 SAMPLE-SPECIFIC PROPERTY

We further analyze the generated filters to show that those filters are sample-specific. We take CI-FAR10 dataset as an example. The model used here is the same trained model used in the CIFAR10 experiment (Section 4.3). In this experiment, we feed a test image A to the classification network and another different image B to generate filters. In other words, the filters that will be used in the classification network are not generated from A but from B. This time the classification accuracy of the classification network falls to 15.24%, which is nearly the random guess. This accuracy drop demonstrates that the generated filters are sample-specific. Filters generated from one image doesn't work on the other image.

## 5 CONCLUSION

In this paper, we propose to learn to generate filters for convolutional neural networks. The filter generation module transforms features from an autoencoder network to sets of coefficients which are then used to linearly combine base filters in filter repositories. Dynamic filters increase model capacity so that a small model with dynamic filters can also be competitive to a deep model. Evaluation on three tasks show the accuracy improvement brought by our filter generation.

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

Table 4: The structure of the baseline model in MNIST experiment.

| Layer | Kernel/Stride/Padding | Output Size |
|-------|----------------------|-------------|
| Conv+ReLU | 5x5/1/0 | 24x24x5 |
| MaxPool | 2x2/2/0 | 12x12x5 |
| Conv+ReLU | 5x5/1/0 | 8x8x5 |
| MaxPool | 2x2/2/0 | 4x4x5 |
| FC | - | 10 |

Table 5: The structure of the autoencoder in MNIST experiment.

| Layer | Kernel/Stride/Padding | Output Size |
|-------|----------------------|-------------|
| Conv+BN+LReLU | 5x5/1/2 | 28x28x$n_{ae}$ |
| Conv+BN+LReLU | 5x5/2/2 | 14x14x$n_{ae}$ |
| Upsample | - | 28x28x$n_{ae}$ |
| Conv | 5x5/1/2 | 28x28x1 |

# A APPENDIX

## A.1 NETWORK STRUCTURES

In this section, we show the details of the network structures used in our experiments.

When we extract sample-specific features, we directly take the convolution feature maps (before LReLU layer) from the autoencoder network as input and feed them to the dimension reduction network. The entire process of sample-specific feature extraction is split into the autoencoder network and the dimension reduction network for the purpose of plain and straightforward illustration.

### A.1.1 NETWORKS FOR MNIST CLASSIFICATION TASK

The networks used in the MNIST experiment are shown from Table 4 to Table 7.

### A.1.2 NETWORKS FOR MTFL FACIAL LANDMARK DETECTION TASK

The networks used in the MTFL experiment are shown from Table 8 to Table 14.

### A.1.3 NETWORKS FOR CIFAR10 CLASSIFICATION TASK

The networks used in the CIFAR10 experiment are shown from Table 15 to Table 20.

## A.2 IMAGE SAMPLES FROM DATASET *D-Align* AND DATASET *D-Rot*

Table 6: Dimension reduction network for the feature maps from 1st layer of the autoencoder.

| Layer | Kernel/Stride/Padding | Output Size |
|---|---|---|
| Conv+BN+LReLU | 4x4/4/0 | $7x7xn_{ae}$ |
| Conv+BN+LReLU | 3x3/3/1 | $3x3xn_{ae}$ |
| Conv+BN+LReLU | 3x3/1/0 | $n_{ae}$ |
| FC1+LReLU | - | $n_{ae}$ |

Table 7: Dimension reduction network for the feature maps from the 2nd layer of the autoencoder.

| Layer | Kernel/Stride/Padding | Output Size |
|---|---|---|
| Conv+BN+LReLU | 2x2/2/0 | $7x7xn_{ae}$ |
| Conv+BN+LReLU | 3x3/3/1 | $3x3xn_{ae}$ |
| Conv+BN+LReLU | 3x3/1/0 | $n_{ae}$ |
| FC1+LReLU | - | $n_{ae}$ |

Table 8: The baseline model in MTFL experiment.

| Layer | Kernel/Stride/Padding | Output Size |
|---|---|---|
| Conv+BN+LReLU | 3x3/1/0 | $64x64xn_{ae}$ |
| Conv+BN+LReLU | 3x3/2/0 | $32x32xn_{ae}$ |
| Conv+BN+LReLU | 3x3/2/0 | $16x16xn_{ae}$ |
| Conv+BN+LReLU | 3x3/2/0 | $8x8xn_{ae}$ |
| Conv2+BN+LReLU | 3x3/2/0 | $4x4xn_{ae}$ |
| Upsample | - | $8x8xn_{ae}$ |
| Conv+BN+LReLU | 3x3/1/0 | $8x8xn_{ae}$ |
| Upsample | - | $16x16xn_{ae}$ |
| Conv+BN+LReLU | 3x3/1/0 | $16x16xn_{ae}$ |
| Upsample | - | $32x32xn_{ae}$ |
| Conv+BN+LReLU | 3x3/1/0 | $32x32xn_{ae}$ |
| Upsample | - | $64x64xn_{ae}$ |
| Conv+BN+LReLU | 3x3/1/0 | $64x64xn_{ae}$ |
| Conv | 3x3/1/0 | 64x64x5 |

Table 9: The structure of the autoencoder network in MTFL experiment.

| Layer | Kernel/Stride/Padding | Output Size |
|---|---|---|
| Conv+BN+LReLU | 3x3/1/0 | 64x64x64 |
| Conv+BN+LReLU | 3x3/2/0 | 32x32x64 |
| Conv+BN+LReLU | 3x3/2/0 | 16x16x96 |
| Conv+BN+LReLU | 3x3/2/0 | 8x8x96 |
| Conv+BN+LReLU | 3x3/2/0 | 4x4x128 |
| Upsample | - | 8x8x128 |
| Conv+BN+ReLU | 3x3/1/0 | 8x8x96 |
| Upsample | - | 16x16x96 |
| Conv+BN+ReLU | 3x3/1/0 | 16x16x96 |
| Upsample | - | 32x32x96 |
| Conv+BN+ReLU | 3x3/1/0 | 32x32x64 |
| Upsample | - | 64x64x64 |
| Conv+BN+ReLU | 3x3/1/0 | 64x64x64 |
| Conv | 3x3/1/0 | 64x64x5 |

Table 10: Dimension reduction network for the feature maps from the 1st layer of the autoencoder network.

| Layer | Kernel/Stride/Padding | Output Size |
|---|---|---|
| Conv+BN+LReLU | 4x4/4/0 | 16x16x64 |
| Conv+BN+LReLU | 4x4/4/0 | 4x4x64 |
| Conv+BN+LReLU | 4x4/1/0 | 64 |
| FC1+LReLU | - | 64 |

Table 11: Dimension reduction network for the feature maps from the 2nd layer of the autoencoder network.

| Layer | Kernel/Stride/Padding | Output Size |
|---|---|---|
| Conv+BN+LReLU | 4x4/4/0 | 8x8x64 |
| Conv+BN+LReLU | 4x4/4/0 | 2x2x64 |
| Conv+BN+LReLU | 2x2/1/0 | 64 |
| FC+LReLU | - | 64 |

Table 12: Dimension reduction network for the feature maps from the 3rd layer of the autoencoder network.

| Layer | Kernel/Stride/Padding | Output Size |
|---|---|---|
| Conv+BN+LReLU | 4x4/4/0 | 4x4x96 |
| Conv+BN+LReLU | 4x4/1/0 | 96 |
| FC+LReLU | - | 96 |

Table 13: Dimension reduction network for the feature maps from the 4th layer of the autoencoder network.

| Layer | Kernel/Stride/Padding | Output Size |
|---|---|---|
| Conv+BN+LReLU | 4x4/4/0 | 2x2x96 |
| Conv+BN+LReLU | 2x2/1/0 | 96 |
| FC+LReLU | - | 96 |

Table 14: Dimension reduction network for the feature maps from the 5th layer of the autoencoder network.

| Layer | Kernel/Stride/Padding | Output Size |
|---|---|---|
| Conv+BN+LReLU | 4x4/1/0 | 128 |
| FC+LReLU | - | 128 |

Table 15: The structure of the baseline model in CIFAR10 experiment.

| Layer | Kernel/Stride/Padding | Output Size |
|---|---|---|
| Conv+BN+ReLU | 3x3/1/0 | 32x32x64 |
| Conv+BN+ReLU | 3x3/2/0 | 16x16x128 |
| Conv+BN+ReLU | 3x3/2/0 | 8x8x256 |
| Conv+BN+ReLU | 3x3/2/0 | 4x4x256 |
| AvgPool | 4x4/1/0 | 256 |
| FC | - | 10 |

Table 16: The structure of the autoencoder network in CIFAR10 experiment.

| Layer | Kernel/Stride/Padding | Output Size |
|---|---|---|
| Conv+BN+ReLU | 3x3/1/0 | 32x32x64 |
| Conv+BN+ReLU | 3x3/2/0 | 16x16x96 |
| Conv+BN+ReLU | 3x3/2/0 | 8x8x128 |
| Conv+BN+ReLU | 3x3/2/0 | 4x4x128 |
| Upsample | - | 8x8x128 |
| Conv+BN+ReLU | 3x3/1/0 | 8x8x128 |
| Upsample | - | 16x16x128 |
| Conv+BN+ReLU | 3x3/1/0 | 16x16x96 |
| Upsample | - | 32x32x96 |
| Conv+BN+ReLU | 3x3/1/0 | 32x32x96 |
| Conv | 3x3/1/0 | 64x64x3 |

Table 17: Dimension reduction network for the feature maps from the 1st layer of the autoencoder network.

| Layer | Kernel/Stride/Padding | Output Size |
|---|---|---|
| Conv+BN+LReLU | 4x4/4/0 | 8x8x64 |
| Conv+BN+LReLU | 4x4/4/0 | 2x2x64 |
| Conv+BN+LReLU | 2x2/1/0 | 64 |
| FC+LReLU | - | 64 |

Table 18: Dimension reduction network for the feature maps from the 2nd layer of the autoencoder network.

| Layer | Kernel/Stride/Padding | Output Size |
|---|---|---|
| Conv+BN+LReLU | 4x4/4/0 | 4x4x96 |
| Conv+BN+LReLU | 4x4/1/0 | 96 |
| FC+LReLU | - | 96 |

Table 19: Dimension reduction network for the feature maps from the 3rd layer of the autoencoder network.

| Layer | Kernel/Stride/Padding | Output Size |
|---|---|---|
| Conv+BN+LReLU | 4x4/4/0 | 2x2x128 |
| Conv+BN+LReLU | 2x2/1/0 | 128 |
| FC+LReLU | - | 128 |

Table 20: Dimension reduction network for the feature maps from the 4th layer of the autoencoder network.

| Layer | Kernel/Stride/Padding | Output Size |
|---|---|---|
| Conv+BN+LReLU | 4x4/1/0 | 128 |
| FC+LReLU | - | 128 |

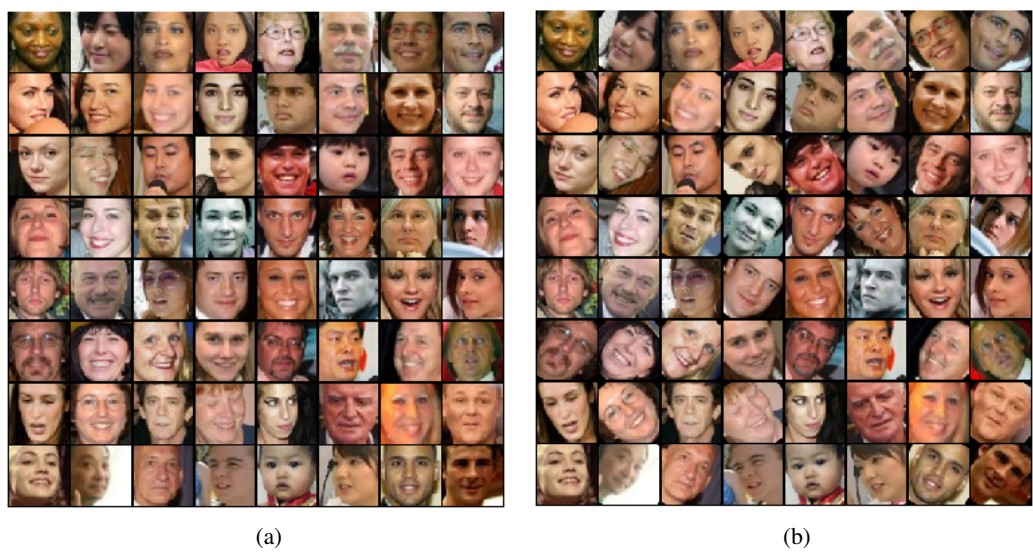

(a)                                                          (b)

Figure 6: (a) Image samples from dataset *D-Align*. (b) Image samples from dataset *D-Rot*.

