# OpenReview forum: "Learning to Generate Filters for Convolutional Neural Networks"
_ICLR.cc/2018/Conference — Reject_

### Official Review · AnonReviewer1 · 2017-11-25
**Reject - little novelty and weak experiments**

**Rating:** 4
**Confidence:** 4

**Review:**

The authors propose an approach to dynamically generating filters in a CNN based on the input image. The filters are generated as linear combinations of a basis set of filters, based on features extracted by an auto-encoder. The authors test the approach on recognition tasks on three datasets: MNIST, MTFL (facial landmarks) and CIFAR10, and show a small improvement over baselines without dynamic filters.

Pros:
1) I have not seen this exact approach proposed before.
2) There method is evaluated on three datasets and two tasks: classification and facial landmark detection.

Cons:
1) The authors are not the first to propose dynamically generating filters, and they clearly mention that the work of De Brabandere et al. is closely related. Yet, there is no comparison to other methods for dynamic weight generation.
2) Related to that, there is no ablation study, so it is unclear if the authors’ contributions are useful. I appreciate the analysis in Tables 1 and 2, but this is not sufficient. Why the need for the autoencoder - why can’t the whole network be trained end-to-end on the goal task? Why generate filters as linear combination - is this just for computational reasons, or also accuracy? This should be analyzed empirically.
3) The experiments are somewhat substandard:
- On MNIST the authors use a tiny poorly-performance network, and it is no surprise that one can beat it with a bigger dynamic filter network.
- The MTFL experiments look most convincing (although this might be because I am not familiar with SoTA on the dataset), but still there is no control for the number of parameters, and the performance improvements are not huge
- On CIFAR10 - there is a marginal improvement in performance, which, as the authors admit, can also be reached by using a deeper model. The baseline models are far from SoTA - the authors should look at more modern architecture such as AllCNN (not particularly new or good, but very simple), ResNet, wide ResNet, DenseNet, etc.

As a comment, I don’t think classification is a good task for showcasing such an architecture - classification is already working extremely well. Many other tasks - for instance, detection, tracking, few-shot learning - seem much more promising.

To conclude, the authors propose a new approach to learning convolutional networks with dynamic input-conditioned filters. Unfortunately, the authors fail to demonstrate the value of the proposed method. I therefore recommend rejection.

---

### Official Review · AnonReviewer3 · 2017-11-28
**Interesting neural network architecture; experiments can be stronger**

**Rating:** 5
**Confidence:** 4

**Review:**

This paper proposes a two-pathway neural network architecture. One pathway is an autoencoder that extracts image features from different layers. The other pathway consists of convolutional layers to solve a supervised task. The kernels of these convolutional layers are generated dynamically based on the autoencoder features of the corresponding layers. Directly mapping the autoencoder features to the convolutional kernels requires a very large matrix multiplication. As a workaround, the proposed method learns a dictionary of base kernels and maps the features to the coefficients on the dictionary.

The proposed method is an interesting way of combining an unsupervised learning objective and a supervised one.

While the idea is interesting, the experiments are a bit weak.
For MNIST (Table 1), only epoch 1 and epoch 20 results are reported. However, the results of a converged model (train for more epochs) are more meaningful.
For Cifar-10 (Figure 4b), the final accuracy is less than 90%, which is several percentages lower than the state-of-the-art method.
For MTFL, I am not sure how significant the final results are. It seems a more commonly used recent protocol is to train on MTFL and test on AFLW.
In general, the experiments are under controlled settings and are encouraging. However, promising results for comparing with the state-of-the-art methods are necessary for showing the practical importance of the proposed method.

A minor point: it is a bit unnatural to call the proposed method “baseline” ...

If the model is trained in an end-to-end manner. It will be helpful to perform ablative studies on how critical the reconstruction loss is (Note that the two pathway can be possibly trained using a single supervised objective function).

It will be interesting to see if the proposed model is useful for semi-supervised learning.

A paper that may be related regarding dynamic filters:
Image Question Answering using Convolutional Neural Network with Dynamic Parameter Prediction

Some paper that may be related regarding combine supervised and unsupervised learning:
Stacked What-Where Auto-encoders
Semi-Supervised Learning with Ladder Networks
Augmenting Supervised Neural Networks with Unsupervised Objectives for Large-Scale Image Classification

---

### Official Review · AnonReviewer2 · 2017-11-28

**Rating:** 4
**Confidence:** 5

**Review:**

This paper explores learning dynamic filters for CNNs. The filters are generated by using the features of an autoencoder on the input image, and linearly combining a set of base filters for each layer. This addresses an interesting problem which has been looked at a lot before, but with some small new parts. There is a lot of prior work in this area that should be cited in the area of dynamic filters and steerable filters. There are also parallels to ladder networks that should be highlighted.

The results indicate improvement over baselines, however baselines are not strong baselines.
A key question is what happens when this method is combined with VGG11 which the authors train as a baseline?
What is the effect of the reconstruction loss? Can it be removed? There should be some ablation study here.
Figure 5 is unclear what is being displayed, there are no labels.

Overall I would advise the authors to address these questions and suggest this as a paper suitable for a workshop submission.

---

### Public Comment · ~Joern-Henrik_Jacobsen1 · 2017-11-09
**Reference**

Nice work, you might be interested in our recent paper on Dynamic Filter Networks with alternative bases: https://arxiv.org/abs/1706.00598

---

> ### Author Response · Authors · 2017-11-10
> **Reference**
>
> Thank you for your comment. I just read your paper. It is very interesting. I will cite your work in the next version.

---

### Decision · Program_Chairs · 2018-01-29
**ICLR 2018 Conference Acceptance Decision**

**Decision:**

Reject

**Comment:**

The paper proposes a method for learning convolutional networks with dynamic input-conditioned filters. There are several prior work along this idea, but there is no comparison agaist them. Overall, experimental results are not convincing enough.